# Unintended consequences of combating desertification in China

Xunming Wang[1,2], Quansheng Ge[1,2], Xin Geng[1,2], Zhaosheng Wang[1], Lei Gao[3], Brett A. Bryan[4], Shengqian Chen[5], Yanan Su[6], Diwen Cai[1], Jiansheng Ye[7], Jimin Sun[8], Huayu Lu[9], Huizheng Che[10], Hong Cheng[11], Hongyan Liu[12], Baoli Liu[13,14], Zhibao Dong[15], Shixiong Cao[16], Ting Hua[17], Siyu Chen[18], Fubao Sun[19], Geping Luo[19], Zhenting Wang[20], Shi Hu[1], Duanyang Xu[1], Mingxing Chen[1], Danfeng Li[1], Fa Liu[1], Xinliang Xu[1], Dongmei Han[1], Yang Zheng[1], Feiyan Xiao[1], Xiaobin Li[1], Ping Wang[1] & Fahu Chen[5,6] ✉

Since the early 2000s, China has carried out extensive "grain-for-green" and grazing exclusion practices to combat desertification in the desertification-prone region (DPR). However, the environmental and socioeconomic impacts of these practices remain unclear. We quantify and compare the changes in fractional vegetation cover (FVC) with economic and population data in the DPR before and after the implementation of these environmental programmes. Here we show that climatic change and $CO_2$ fertilization are relatively strong drivers of vegetation rehabilitation from 2001-2020 in the DPR, and the declines in the direct incomes of farmers and herders caused by ecological practices exceed the subsidies provided by governments. To minimize economic hardship, enhance food security, and improve the returns on policy investments in the DPR, China needs to adapt its environmental programmes to address the potential impacts of future climate change and create positive synergies to combat desertification and improve the economy in this region.

China's desertification-prone region (DPR) stretches from central Asia in the western direction to north-eastern China in the eastern direction, covering an area of more than 1.2 million km² (see Methods and Supplementary Note 1). At present, more than 60% of the DPR is managed using traditional pastoral and agricultural systems, and impacts of desertification on farming and grazing affect the lives of over 47.9 million people[1,2].

Due to the potential significant effects of desertification on China's ecology and food security, the Chinese government has carried out several desertification-combating actions to improve vegetation condition[3,4], and since the early 2000s, more major countermeasures involving widespread "grain-for-green" and grazing exclusion practices were implemented[5] (Supplementary Note 2). These activities have been implemented under a number of environmental programmes,

and are supported by laws and regulations such as the Grassland Law[6,7]. Since 2002, direct investments in "grain-for-green" and grazing exclusion practices in the DPR and adjacent areas have exceeded 780 billion RMB (~112 billion USD) (Supplementary Table 1), and according to government planning, this investment may soon be strengthened (Supplementary Table 2). However, the benefits of these ambitious practices and investments for combating desertification remains unclear, and few studies have assessed their broader impacts on sustainability[8–10].

Here we quantify vegetation responses to climate variability using statistical models combined with multisource remote sensing methods, analyze the impacts of desertification-combating actions on vegetation trends as well as on agriculture and livestock production based on differences between reality and the without-practice

**Fig. 1 | The contributions of desertification-combating practices to vegetation restoration.** Spatial and statistical distributions of the contributions (%) of "grain-for-green" **a**, **b** and grazing exclusion **c**, **d** practices to vegetation restoration since 2000 and 2003, respectively. The x-axis in **b**, **d** corresponds to the contributions in **a**, **c**. The scope of the desertification-prone region is marked in panels **a** and **c** with gray. For each pixel in **a**, **c**, the average fractional vegetation cover (FVC) trend resulting from climate change and $CO_2$ fertilization (natural FVC trends) is derived by satellite and multiple linear regression simulations, while the contributions of intervention practices are estimated based on the average difference between the natural FVC trend and the actual trend involving practices implementation. $Ave.\alpha$

and $Ave.\beta$ in **b**, **d** are the integrated contributions of the "grain-for-green" and grazing exclusion practices, respectively. These values are calculated based on area-weighted statistics of the pixel-level contributions in **a** and **c** as follows: $Ave.\alpha$ (or $\beta$) = $\Sigma$ [$\alpha$ (or $\beta$)$_i$·area$_i$]/$\Sigma$area$_i$, where $\alpha$ ($\beta$)$_i$ is the vegetation restoration contribution of the two practices in $i$-th pixel involved, and $area_i$ is the area of the $i$-th pixel. Noted that only regions with significant trends (passed the Mann-Kendall test at the 95% significance level) were considered in area-weighted statistics and rest were shown by the dotted box filled with gray in **b**, **d**. See Methods for more details about the identification of pixels involved in the "grain-for-green" and grazing exclusion practices, and about the calculation of pixel-level contributions.

hypothesis, and forecast vegetation growth under future climate scenarios using robust stepwise multiple linear regression models (see Methods). These analyses are performed to support the comprehensive assessment of the environmental and economic impacts of the "grain-for-green" and grazing exclusion practices implemented in the DPR of China over the past 20 years. The results suggest opportunities for adapting China's desertification combating practices and creating positive synergies to benefit the livelihoods[11] and food security of those living in the DPR and for improving the ecological environment, thereby contributing to several UN sustainable development goals (SDGs)[12].

## Results

### Impacts of desertification-combating practices on vegetation trends

Although extensive "grain-for-green" and grazing exclusion practices have been implemented in China over the past 20 years (2000 to 2020), desertification reversals have been reflected by the fractional vegetation coverage (FVC) increasing in most grasslands and croplands in the DPR since 1982 (Supplementary Fig. 1; Supplementary Table 3; see Methods and Supplementary Note 3 for the FVC dataset we constructed herein), suggesting that environmental factors triggered vegetation recovery earlier than government interventions. Here, we applied a statistical framework to identify the contribution of climate change and the two types of intervention practices to vegetation restoration in the DPR by simulating and detecting satellite-derived natural FVC trends in grasslands and croplands without these desertification-combating practices, and further isolating the practices effects of such practices on actual FVC variations (see Methods). Our analyses reveal that, compared to adjacent lands (see Methods), 63% of the restored land involved in the "grain-for-green" practices made positive contributions to FVC increases since 2000, while 14% showed negative effects, as vegetation restoration cannot offset the consequences of crop removal (Fig. 1a); the average contribution of the "grain-for-green" practices to FVC increases was estimated at −1.06% in the DPR (Fig. 1b). In addition, our results also show that although vegetation restoration occurred in 9.44% of grasslands in the DPR with grazing exclusion implementations (Supplementary Table 3), after excluding the FVC trends triggered by climate change and $CO_2$

fertilization (see Methods; Fig. 2), areas with net FVC increases triggered by grazing exclusion comprised only 20.45% of these regions (Fig. 1c). Further area-weighted statistical analyses show that the average contribution of grazing exclusion practices to vegetation restoration in the DPR was 13.40% (Fig. 1d), implying that even without grazing exclusion practices, the natural resilience of vegetation to adapt to the original grazing intensity is still considerable. In total, the joint contribution of both analyzed ecological practices to FVC increases in the DPR was only 13.07%.

### Impacts of desertification-combating practices on grain and meat production

Our results also show that the desertification-combating practices enacted in the DPR may jeopardize the food security of China. Historically, several areas in the DPR, including Horqin, Ordos, Tarim and the Hexi Corridor (Supplementary Fig. 2), have been the main areas for meat and grain production in China[13] and are thus crucial for China's food supply. However, the desertification combating practices have resulted in reductions in available farmlands and grasslands (Supplementary Fig. 3) and, consequentially, have reduced both grain and meat production in the DPR (Supplementary Fig. 4). Because the "grain-for-green" and grazing exclusion practices were extensively launched with unambiguous impacts on available lands after 2000 and 2010, respectively (Supplementary Note 2), by using the areas of farmlands in 2000 and of grasslands in 2010, we estimated that the expected grain and meat production in China's DPR would be 28.2 million tonnes of grain and 106.0 million sheep (~2.1 million tonnes of meat) in 2020 (Fig. 3; Supplementary Table 4; see Methods). However, due to the restrictions placed on farming and grazing activities by the "grain-for-green" and grazing exclusion practices, compared to the expected yields, the mean costs in terms of foregone grain and meat production in the DPR were 13.4% and 24.2%, respectively, from 2001 to 2020 (Fig. 3; Supplementary Table 4; see Methods). Based on the basic requirements suggested by Chinese government[14] of 400 kilograms of grain per capita and 21 kilograms of meat per capita[15], these results mean that the present outputs of grain and meat in the DPR could maintain population sizes of only 59.9 million, far below the expected population of 70.6 million people in 2020.

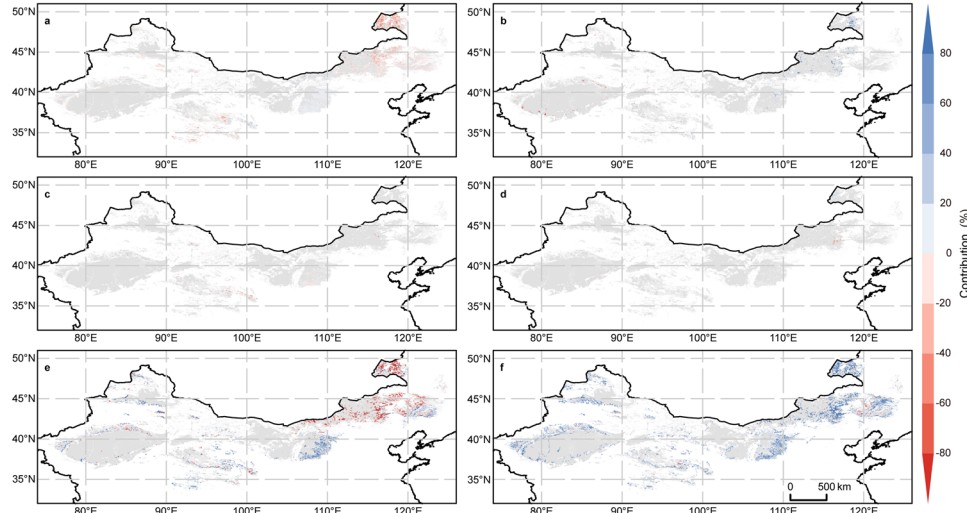

**Fig. 2 | The contributions of natural factors to vegetation restoration.** Contributions of the precipitation **a**, temperature **b**, solar radiation intensity **c**, near-surface wind speed **d**, and atmospheric $CO_2$ concentration **e** to vegetation recovery in the desertification-prone region (DPR) and their total contribution **f** from 2000 to 2018. The scope of the DPR is marked in **a**–**f** with gray. These contributions were estimated based on the factor coefficients of the stepwise multiple linear regression equation (see Methods). For each pixel, only factors with significant effects (those that passed the F-test at the 95% significance level) were considered.

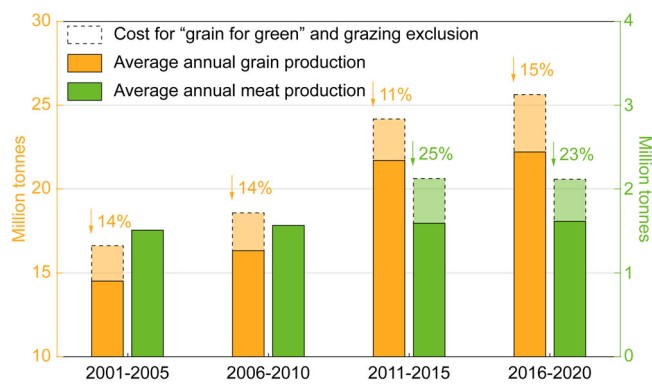

**Fig. 3 | Impacts of desertification-combating practices on grain and meat production.** Grain and meat production losses (shown by the dotted box in the figure) caused by restrictions on available lands put in place by the "grain-for-green" and grazing exclusion practices in the desertification-prone region. Because these two practices have been extensively launched since 2001 and 2011, respectively (Supplementary Note 2), the grain and meat production losses were estimated for the simultaneous periods. More details are provided in the Methods section.

## Impacts of desertification-combating practices on the direct incomes of farmers and herders

Another impact of the "grain for green" and grazing exclusion desertification-combating practices may be the exacerbation of poverty in the DPR. In 2020, the gross domestic product (GDP) in this region was ~1,092 billion Yuan (~153 billion USD), of which 8.6% was from farming and grazing activities (Supplementary Table 5). Moreover, 45.4% of the household disposable income of local farmers and herders is derived directly from farming and grazing activities (Supplementary Table 5). At present, the direct compensation for desertification-combating practices in the DPR and the adjacent regions is only ~5.70 billion RMB (~0.80 billion USD), representing only ~6.10% of the direct income of the local farmers and herders (Supplementary Table 6); meanwhile, the direct income of farmers and herders in the region (~75.1 billion RMB yr$^{-1}$ and ~10.5 billion USD yr$^{-1}$) decreased by 15.0% from 2001 to 2020 compared to expectations

(~88.4 billion RMB yr$^{-1}$ and ~12.4 billion USD yr$^{-1}$) as a result of "grain-for-green" and grazing exclusion practices (Supplementary Table 4; see Methods). With the extension of these two practices, the substantial economic losses of farmers and herders may further aggravate the region's impoverishment.

## Discussion

By enacting a series of ecological programmes to restore regional vegetation coverage[16–18], China has played a vital role in guiding the ultimate realization of the SDGs[5], and the related actions are of inspiration in mitigating global warming, achieving carbon neutrality[19], and preventing and controlling land degradation[20]. However, since the 1980s, vegetation greenness has been increasing globally, and this increase has been argued to be mainly driven by climate change, agricultural progress, and $CO_2$ fertilization[21,22]. Over the past 40 years, the DPR of China has experienced widespread warming and wetting, with overall increasing trends of 2.43 mm per decade in precipitation and 0.37 °C per decade in temperature (Supplementary Fig. 5); together with $CO_2$ fertilization effects, these trends have dominated the FVC increase in some grasslands and most farmlands of the region (Fig. 2). Following the forecasts obtained the Coupled Model Intercomparison Project (CMIP6), this warming and wetting climate trend may continue until 2050 (Supplementary Fig. 5) and is expected to improve vegetation restoration in at least 68.12%, 65.06%, and 56.29% of farmlands, forests, and grasslands in the DPR, respectively, as well as to result in an 8.17% increase in FVC across the whole DPR (Fig. 4; Supplementary Table 7). Therefore, although extensive cultivation with population migrating into the DPR in historical periods has triggered dust storms in northern China[23], the current desertification-combating practices in the DPR of China seem to have been excessively implemented.

At present, although unsustainable land use practices still threaten to land and environmental degradation[24,25], the active management of grazing and farming plays a major role in ecosystem health and sustainability[26]. Unlike other regions of the world[27], the "grain-for-green" and grazing exclusion practices currently carried out in China may not be highly effective against desertification. For instance, the "grain-for-green" practices has not met the expected vegetation restoration magnitude in the short term (Fig. 1a), and the fences used

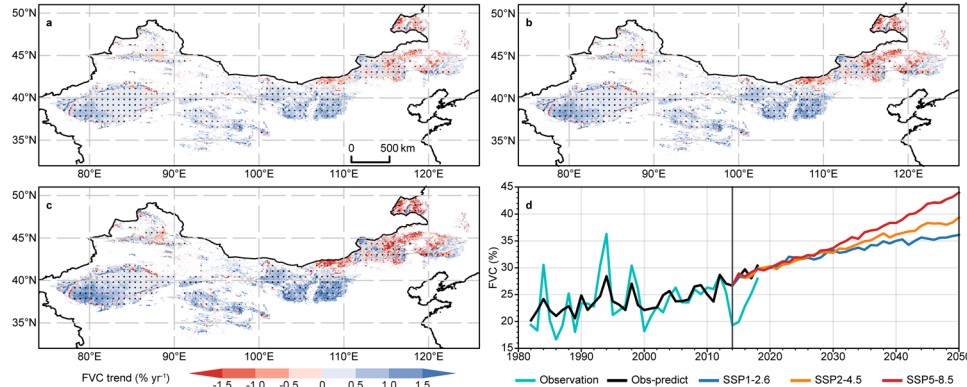

**Fig. 4 | Future projections of fractional vegetation cover (FVC) trends.** FVC trends estimated by the stepwise multiple linear regression (SMLR) model from 2015 to 2050 under different shared socioeconomic pathway (SSP) and representative concentration pathway (RCP) scenarios (i.e., SSP1-2.6 **a**, SSP2-4.5 **b**, and SSP5-8.5 **c**) from the Coupled Model Intercomparison Project Phase 6 (CMIP6) model experiments and their corresponding trend lines **d**. The stippling in **a**–**c** marks regions with significant trends (passed the Mann-Kendall test at the 95% significance level). The line referred to as Obs-predict in **d** was estimated based on the SMLR model and historical environmental factors. All scenario simulations were modified by removing the errors between the simulated and observed data during the reference period of 1982–2014 (Supplementary Note 4), and the differences among the three scenarios are shown in **d** since 2014.

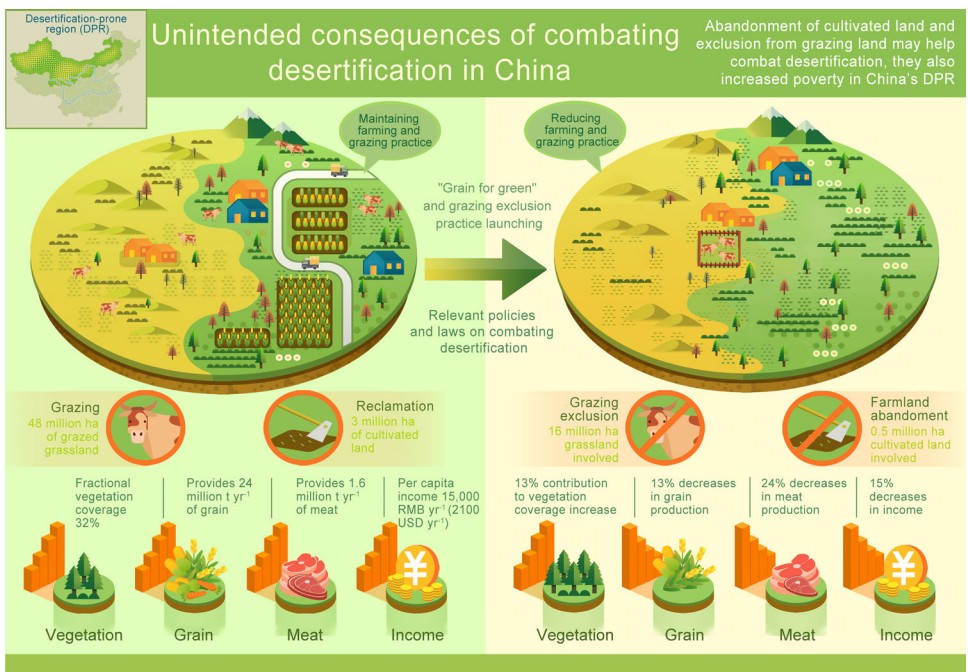

**Fig. 5 | Impacts of the "grain-for-green" and grazing exclusion practices in the desertification-prone region.** At present, 3.2 million and 48.3 million hectares of land are used for farming and grazing in the desertification-prone region, respectively. These available farmland and grassland provided 24 million tonnes of grain and 1.6 million tonnes of meat in 2020 (Supplementary Fig. 7) and generated 15,000 RMB (2,100 USD) of disposable income per capita (Supplementary Table 5). However, the practices of "grain-for-green" and grazing exclusion practices have resulted in decreases of 13%, 24%, and 15% in grain production, meat production, and incomes of framers and herdsmen, respectively, and have contributed only 13% to vegetation restoration (Fig. 3; Supplementary Table 4).

for grazing exclusion may hinder wildlife migration, increase grazing pressure in these unfenced areas and cause additional habitat destruction[28]. In comparison, human-interposed irrigation and ploughing systems may be more appropriate for imperceptibly optimizing the physical and chemical properties of soils and, subsequently, may increase the productivity of vegetation[29,30]; in addition, proper grazing is conducive to extending the growth cycle of vegetation and increasing grass yields[31,32]. Additionally, our analyzed results show that the land areas of the current DPR used for farming and grazing are only 3.2 million and 48.3 million hectares, respectively (Supplementary Fig. 6). The low proportion and scattered spatial distributions of these farming and grazing lands developed in the DPR do not trigger serious

desertification issues; therefore, desertification-combating practices such as the "grain-for-green" and grazing exclusion practices might be suboptimal with regards to rational land management in this region.

The "grain-for-green" and grazing exclusion practices were designed to reduce desertification and improve regional ecosystem stability[5]. However, due to their impacts on vegetation, agriculture, husbandry and the incomes of farmers and herders in the DPR (Fig. 5), China's current desertification-combating programmes need to be improved. According to our results, most of the DPR has long benefited from the positive effects of climate change, $CO_2$ fertilization and agricultural management rather than of government actions (Figs. 1 and 2). Therefore, such land use restrictions in these areas are

nonurgent, and are even suggested to be terminated[33,34]. In addition, in some parts of the DPR, the "grain-for-green" and grazing exclusion practices may be too primitive to combat desertification (Figs. 1 and 4), and more practices, such as the establishment of farmland shelterbelts[35], water conservation in agriculture[36], and the implementation of rotational grazing[37], may be more suitable for combating desertification in these regions. In addition, although subsides for the income losses of farmers and herders caused by ecological programmes benefit the livelihood of the affected people[38], the practices of these strategies are unwarranted and unproven[39] and may enlarge income inequalities within local communities[40]. At present, direct subsidy improvements and the creation of a more transparent fund management system may be expected to maximize the benefits of farmers and herders, but these programmes are unsustainable under the current financial pressures of China's governments[41]. During the policy-formulation processes desertification-combating practices, policy-makers should maximize the benefits of both humans and the ecological environment and create positive synergies to increase farmer and herder incomes, combat desertification and improve the ecological environment in the DPR of China.

## Methods

### Determining the DPR boundary

We used the spatial distribution of the DPR produced by the Cold and Arid Regions Environmental and Engineering Research Institute (CAREERI) of the Chinese Academy of Sciences in 2000[42–45], together with the Desert Distribution Map of China produced by the Institute of Glaciology, Frozen Soil and Desert, Institute of Geography Research, Chinese Academy of Sciences[46,47]. The DPR is mainly located in the arid, semiarid, and semihumid regions of China; however, the Tibet District and several desertified areas scattered in southern China are not included. In addition, the DPR discussed in this study does not include Gobi deserts or salinized lands, as explained in Supplementary Note 1. Using thermal mapper (TM) images taken in 2000 as the main source and combining these images with maps of the local topography (1:100,000), soil, vegetation, geology, and geomorphology, we compiled the spatial distributions of these variables in the DPR (Supplementary Fig. 2). The area of the DPR in China was estimated to be $122.3 \times 10^4 \, \text{km}^2$, and, using the definitions of the geographical regions in China[43,48], the DPR was divided into the following subregions: Hunlun Buir, Nunkiang, Horqin, Otindag, Erdos, Ala Shan & Hexi Corridor, Qinghai, Tarim, and Junggar (Supplementary Fig. 2).

### Land use data

Thirty-meter-resolution land use data representing the 1980s, 1990, 1995, 2000, 2005, 2010, 2015, and 2020 in the DPR were obtained from the Multi-Period Land Use Land Cover Remote Sensing Monitoring Dataset for China (CNLUCC) from the Data Center for Resources and Environmental Sciences (http://www.resdc.cn) and used to determine land use changes involved in the "grain-for-green" and grazing exclusion practices in the DPR. The CNLUCC data were generated by visual interpretations of satellite imagery; the 1980 data were based on Landsat-MSS, the 1990–2010 data were based on Landsat-TM/ETM, and the 2015–2020 data were based on Landsat 8[49]. The accuracy of the interpreted data was determined by several field surveys performed over the years of study, and the results show that the average classification accuracy of the 25 subcategories exceeded 90%[50], indicating that the CNLUCC land use data can be used as reliable base maps for subsequent analyses. See Supplementary Note 5 for details of these surveys.

### Economic data

To assess the production and poverty statuses, we collected population, disposable income, GDP, grain production, planted area, and livestock production data. First, we obtained county-level socioeconomic data from the China Database on Country-level Agricultural and Rural Indicators (http://tongji.cnki.net/kns55/Navi/NaviDefault.aspx); these data were collected by each county's statistical station and reported to upper-level statistical bureaus. We used linear regression to replace outliers and missing values in the series, extrapolated the 2020 statistical data based on the linear trend from 1982-2019, and converted livestock data into standard sheep units[51]. A cross-validation analysis was then performed to validate the accuracy of the imputations[52]. Second, we extracted the population, disposable income and GDP data based on the spatial distribution of the population density (1.6% deviations compared to the country-level database) from the Resources and Environmental Science Data Center of the Chinese Academy of Sciences (RESDC, http://www.resdc.cn). The grain production, planted area, and livestock production data were calculated using the converted county-level production per unit area grassland and farmland from the country-level database and the total area of grasslands and farmlands from CNLUCC. In addition, we calculated government subsidies to DPR farmers and herders based on the areas affected by desertification-combating practices (as clarified below) and the total payments reported in official literature (Supplementary Table 6). The accuracy of these downscaled data can be ensured based on the robustness of the CNLUCC data.

### Fractional vegetation coverage (FVC) data

To assess the vegetation changes in the DPR, we reproduced a 250-m-resolution FVC dataset (CD FVC) using the improved pixel bipartite model[53,54] based on a 250-m-resolution constructed normalized difference vegetation index (NDVI) dataset (CD NDVI) for 1982-2018 (for more details about the CD NDVI dataset, see Supplementary Note 3):

$$\text{CD FVC} = \frac{\text{NDVI} - \text{NDVI}_{min}}{\text{NDVI}_{max} - \text{NDVI}_{min}} \quad (1)$$

$$\text{NDVI}_{min} = \frac{1}{n} \sum_{i=1}^{i=n} \text{NDVI}_{y,min} \quad (2)$$

$$\text{NDVI}_{max} = \frac{1}{n} \sum_{i=1}^{i=n} \text{NDVI}_{y,max} \quad (3)$$

where $\text{NDVI}_{max}$ (or $\text{NDVI}_{y,max}$) and $\text{NDVI}_{min}$ (or $\text{NDVI}_{y,min}$) correspond to the NDVI values representing surfaces with a fully covered dense vegetation and bare soils, respectively. To reduce the uncertainty and randomness involved in determining extreme NDVI values[55], we first selected the NDVI values (NDVI > 0) at the 5% and 95% percentiles from the surface grid values as the annual $\text{NDVI}_{y,min}$ and $\text{NDVI}_{y,max}$ for each year, respectively, and then calculated the multiyear average $\text{NDVI}_{y,min}$ and $\text{NDVI}_{y,max}$ values as the bare soil pixel ($\text{NDVI}_{min}$) and the fully vegetated pixel ($\text{NDVI}_{max}$) in the study area for the period from 1982 to 2018.

We tested the reliability of the results by performing a trend consistency evaluation with published datasets. The results show that annual CD FVC variation trends of the entire region and subregions were all consistent with those of the Blended Vegetation Health (VH), Global Inventory Modeling and Mapping Studies (GIMMS3g), Satellite Pour l'Observation de la Terre (SPOT), and Moderate-resolution Imaging Spectroradiometer (MODIS) FVC datasets (Supplementary Fig. 8; Supplementary Table 8; Supplementary Table 9), suggesting that the reproduced dataset can effectively depict the desertification variation in the DPR. In addition, during data processing we found that the updated CD FVC data in 2019 and 2020 had some quality flaws. Therefore, in this study, we used only the CD FVC data spanning from 1982 to 2018.

## Climate observation data

The maximum temperature (TMX), minimum temperature (TMN), precipitation (PRE), solar radiation (SR) and mean wind speed (WS) data recorded from 1982 to 2018 were used as control variables in the contribution analyses. The Terra-climate monthly dataset was obtained from the University of Idaho's Northwest Knowledge Network (https://climatedataguide.ucar.edu/climate-data/terraclimate-global-high-resolution-gridded-temperature-precipitation-and-other-water) at a spatial resolution of 1/24° (~4 km)[56], and annual temperature (TEM, mean value of TMX and TMN), PRE, SR and WS data were further calculated by averaging the monthly values over the growing season (April to October). In addition, monthly 0.5°-spatial-resolution $CO_2$ concentration data representing the 1982–2018 period were obtained from the work by Meinshausen et al.[57], which was developed for the Coupled Model Intercomparison Project Phase 6 (CMIP6) model experiments (https://greenhousegases.science.unimelb.edu.au/#!/view), and these data were also converted to annual growing season values.

## Climate prediction data

The outputs of 21 commonly used global climate models (GCMs) (Supplementary Table 10) from CMIP6, including the monthly precipitation, air temperature (mean, maximum, and minimum), solar irradiation, and wind and $CO_2$ concentration outputs, from 1982 to 2050 were downloaded from the Earth System Grid Federation (ESGF, https://esgf-node.llnl.gov/search/cmip6/). These data were used to assess the future climate and $CO_2$ variations as well as the impact of these variations on vegetation restoration in the DPR. The time-series of CMIP6 model outputs comprised (1) historical simulations (1980–2014) used to facilitate comparisons with the observed data and (2) ScenarioMIP simulations (2015-2050) used to show future changes in the climate system under different scenarios of shared socio-economic pathway (SSP) and representative concentration pathway (RCP) scenarios[57]. All climate simulations were corrected by removing the errors between the simulated and observed data in the 1982–2018 period. More details regarding these data and methods are provided in Supplementary Note 4.

## Areas covered by "grain-for-green" and grazing exclusion practices

To confirm the areas in the DPR involved in the "grain-for-green" and grazing exclusion practices, we detected the land use changes after 2000. First, the land uses from 1980 to 2000 were employed to determine the reference area of permanent farmlands (pmnt FL; "permanent" means the land use type was consistent during the observation period) before the "grain-for-green" practices were launched, and the restored lands in 2000–2005, 2000–2010, 2000–2015, and 2000–2020, which included areas that were permanent farmlands from 1980 to 2000 but changed to forests, grasslands or unused lands from 2005–2020, were extracted as the areas involved in "grain-for-green" practice. Second, considering that the policies related to grazing exclusion were fully implemented and recorded in each province in China after 2010 (Supplementary Note 2), the area of grassland area related to grazing exclusion at the provincial scale could be refined to the DPR (Supplementary Fig. 3). The relevant analyses were performed using ArcMap 10.6. Noted that the programmes implemented before 2000s such as the Great Green Wall Program (launched in 1978) were not considered in this work although they had influence on vegetation condition[3,4], because they did not link to the issues about ecological and economic trade-offs arising from farmland and grassland restrictions.

## Estimating the contributions of climate change, $CO_2$ fertilization, and "grain-for-green" and grazing exclusion practices to vegetation restoration

A statistical framework was applied to identify the contributions of climate change, $CO_2$ fertilization, and the areas covered by "grain-for-green" and grazing exclusion practices to explaining the FVC variance (Supplementary Fig. 9). For the "grain-for-green"-affected areas, we created a 500-meter buffer zone based on the "Buffer" tool in ArcMap 10.6 for each RL pixel (Supplementary Fig. 9a) in which adjacent restored lands and permanent lands (referred to as pmnt in the following text, including permanent farmlands, permanent forests, and permanent unused lands) with similar environmental characteristics were merged[26]. Without being affected by the "grain-for-green" practices, the adjacent pmnt's FVC change can be considered as a reference for estimating the intervention practice effect on the FVC change of restored lands; therefore, the contribution of "grain for green" practice to the FVC increase ($\alpha$) was identified as follows:

$$\alpha = \frac{FVC_{RL}' - FVC_{pmnt}'}{|FVC_{RL}'|} \times 100\% \quad (4)$$

where $FVC_{RL}'$ and $FVC_{pmnt(c)}'$ are the mean linear trends of restored lands and pmnt pixels after the implementation of "grain-for-green" practice, respectively; these terms are estimated using the least square method and satellite observations. As changes in FVC in permanent grasslands might also be influenced by grazing exclusions, we included only permanent farmlands, permanent forests, and permanent unused lands in the contribution calculation of "grain-for-green" practices.

To evaluate whether grazing exclusion contributed to vegetation changes, we removed the impacts of land use changes; thus, only permanent grasslands were analyzed. In addition, to remove the impacts of environmental factors, including climate change and $CO_2$ fertilization, we divided the study period into two periods (before and after grazing exclusion) according to the years in which grazing exclusion was implemented in different areas (Supplementary Note 2). A regression model was established for the climatic factors and FVC before grazing exclusion, and this model was then used to simulate the natural FVC trend in the period after grazing exclusion. By comparing the difference between the simulated and observed FVC trends, the change in the FVC trend caused by grazing exclusion could be determined (Supplementary Fig. 9b). Due to the wide spatial range and heterogeneities in the study region, spatial differences may occur in the dominant environmental variables driving vegetation growth[58]. Therefore, we applied stepwise multiple linear regression (SMLR) to each pixel and included only the significant environmental variables (those that passed the F test at the 95% significance level). The contribution of grazing exclusion to vegetation change ($\beta$) was calculated using Eqs. 5–10:

$$FVC_{actual,bfr} = b + \sum_{i \in env} a_i x_{bfr,i} + FVC_{res,bfr} \quad (5)$$

$$FVC_{predict,aft} = b + \sum_{i \in env} a_i x_{aft,i} + FVC_{res,aft} \quad (6)$$

$$\overline{FVC_{res,bfr}} = \overline{FVC_{res,aft}} \quad (7)$$

$$FVC_{actual,aft} = FVC_{predict,aft} + FVC_{GE,aft} \quad (8)$$

$$FVC_{actual}' = FVC_{actual,aft} - FVC_{actual,bfr} = FVC_{env}' + FVC_{GE}' \quad (9)$$

$$\beta = \frac{FVC_{GE}'}{FVC_{actual}'} \times 100\% = \frac{FVC_{actual,aft} - FVC_{predict,aft}}{FVC_{actual,aft} - FVC_{actual,bfr}} \times 100\% \quad (10)$$

where $a_i$ is the regression coefficient of environmental variable i; b is the constant term; $x_i$ is the dataset of environmental variable i; and env represents environmental variables, including PRE, TEM, SR, SW and $CO_2$. The env variables used in the regression were determined by

stepwise multiple regression, and PRE is a mandatory variable. The term bfr and aft represent the periods before and after grazing exclusion, respectively, and the breakpoints are set independently for each province with reference to the yearbook. $FVC_{actual}$, $FVC_{predict}$ and $FVC_{res}$ are the observed, predicted, and residual FVC, respectively. The residual term is the FVC change caused by other factors, such as grazing, nitrogen deposition and recovery from natural disturbances[59], that cannot be explained by the regression model; these changes were ignored in the FVC change comparison, as these disturbances remained unchanged before and after grazing exclusion practices were implemented. $FVC_{GE}$ is the FVC change caused by grazing exclusion practices; this term was included in the actual FVC sequence after grazing prohibition. $FVC_{actual}'$, $FVC_{env}'$, and $FVC_{GE}'$ are the observed, climate-induced and grazing exclusion-caused changes in FVC, respectively. Environmental factors are the only driving forces of FVC after grazing exclusion practices are implemented, and the i-th factor's contribution ($\beta_i$) was calculated as follows:

$$\beta_i = \frac{a_i x_{aft,i}'}{FVC_{env}'} \times 100\% \qquad (11)$$

where $x_{aft,i}'$ is the change in the i-th factor. Because the contribution estimates in this study are based on the long-term FVC trend, we used the coefficient of determination ($R^2$, calculated as Eq. 12), mean absolute error (MAE, calculated as Eq. 13) and root mean square error (RMSE, calculated as Eq. 14) to estimate the model errors corresponding to each pixel.

$$R^2 = \frac{\sum_{i=1}^{n}(\hat{y}_i - \bar{y}_i)^2}{\sum_{i=1}^{n}(y_i - \bar{y}_i)^2} \qquad (12)$$

$$MAE = \frac{1}{n}\sum_{i=1}^{n}|\hat{y}_i - y_i| \qquad (13)$$

$$RMSE = \sqrt{\frac{1}{n}\sum_{i=1}^{n}(y_i - \hat{y}_i)^2} \qquad (14)$$

where n is the period length, $y_i$ is the observed FVC, $\bar{y}_i$ is the mean $y_i$ value, and $\hat{y}_i$ is the simulated FVC. In this study, multiple regression models explained 0.53 ($R^2$) of the change in the average in FVC and shown low values of MAE and RMSE across all land-use types (Supplementary Fig. 10), meeting the simulation requirements. All trend and modeling analyses were conducted based on packages of 'scipy', 'numpy', 'pandas', and 'statsmodels' in Python 3.8. With this method and the climate predictions from the CMIP6, we extended the FVC simulation to 2050 in the entire DPR.

### Income and economic losses caused by the "grain-for-green" and grazing exclusion practices

Grain and livestock prices during 1982–2019 were obtained from relevant statistical yearbooks (National Development and Reform Commission of China, 2002–2020; Department of Rural Social-Economic Survey, National Bureau of Statistics, 1983–2020). Various agricultural indices were calculated as follows: grain yield = total grain production/cultivated land area; grain income = grain production × grain price; sheep per unit area = total sheep/grassland area; meat production = meat production per sheep × total number of sheep. The meat production per sheep unit was set to -20 kg[60]. In addition, the calculations of sheep units per unit grassland area, meat production, and grazing income after 2011 were acquired based on the grassland areas, but the areas involved in grazing exclusion were excluded. The decreases in grain production could be directly estimated from decreases in the cultivated land area in the DPR.

## Data availability

All data used in this study are available online. The county-level socioeconomic data were from the China Database on Country-level Agricultural and Rural Indicators (http://tongji.cnki.net/kns55/Navi/NaviDefault.aspx). The Multi-Period Land Use Land Cover Remote Sensing Monitoring Dataset for China (CNLUCC) from the Data Center for Resources and Environmental Sciences (http://www.resdc.cn). The CMIP6 model simulations and the $CO_2$ concentration data were from the Earth System Grid Federation (ESGF, https://esgf-node.llnl.gov/search/cmip6/). The terra-climate monthly dataset was obtained from the University of Idaho's Northwest Knowledge Network (https://climatedataguide.ucar.edu/climate-data/terraclimate-global-high-resolution-gridded-temperature-precipitation-and-other-water). The VH NDVI data were from the Center for Satellite Applications and Research (https://www.star.nesdis.noaa.gov/smcd/emb/vci/VH/vh_ftp.php), and the MODIS NDVI was from the Level-1 and Atmosphere Archive & Distribution System Distributed Active Archive Center (https://ladsweb.modaps.eosdis.nasa.gov/missions-and-measurements/products/MOD13Q1).

## Code availability

All relevant software and packages used for data analyze in this paper are clarified in the "Methods" section. The relevant codes that were used to produce CD FVC dataset are available from the corresponding author on request.

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

## Acknowledgements

We are very grateful to Prof. James F. Reynolds for his critical comments and detailed editing on the draft. This work was supported by National Natural Science Foundation of China (41790421), the Second Tibetan Plateau Scientific Expedition and Research (2019QZKK0305) and Key Frontier Program of Chinese Academy of Sciences (QYZDJ–SSW–DQC043).

## Author contributions

F.C., X.W., and Q.G. designed the study. F.C., X.W., X.G., Z.W. (Zhaosheng Wang), L.G. and B.B. led the interpretation and writing. X.G., Z.W. (Zhaosheng Wang), D.C., T.H., F.S., G.L., Z.W. (Zhenting Wang), S.H., D.X., M.C., D.L., F.L., X.X., D.H., Y.Z., F.X., X.L., P.W., and F.C. led the data analysis work. X.W., Q.G., X.G., and F.C. wrote the manuscript with substantial contributions from S.C. (Shengqian Chen), Y.S., J.Y., J.S., H.L. (Huayu Lu), H.C. (Huizheng Che), H.C. (Hong Cheng), H.L. (Hongyan Liu), B.L., Z.D., and S.C. (Shixiong Cao).

## Competing interests

The authors declare no competing interests.

## Additional information

**Supplementary information** The online version contains ?supplementary material available at https://doi.org/10.1038/s41467-023-36835-z.

[1]Institute of Geographic Sciences and Natural Resources Research, Chinese Academy of Sciences, Beijing, China. [2]College of Resources and Environment, University of Chinese Academy of Sciences, Beijing, China. [3]CSIRO, Waite Campus, Adelaide, South Australia, Australia. [4]School of Life and Environmental Sciences, Deakin University, Melbourne, Victoria, Australia. [5]ALPHA, State Key Laboratory of Tibetan Plateau Earth System, Environment and Resources (TPESER), Institute of Tibetan Plateau Research (ITPCAS), Chinese Academy of Sciences (CAS), Beijing, China. [6]College of Earth and Environmental Sciences, Lanzhou University, Lanzhou, China. [7]School of Life Sciences, Lanzhou University, Lanzhou, China. [8]Key Laboratory of Cenozoic Geology and Environment, Institute of Geology and Geophysics, Chinese Academy of Sciences, Beijing, China. [9]School of Oceanographic and Geographic Sciences, Nanjing University, Nanjing, China. [10]State Key Laboratory of Severe Weather, Institute of Atmospheric Composition, Chinese Academy of Meteorological Sciences, Beijing, China. [11]State Key Laboratory of Earth Surface Processes and Resource Ecology, Beijing Normal University, Beijing, China. [12]College of Urban and Environmental Sciences, Peking University, Beijing, China. [13]School of Geography and the Environment, University of Oxford, Oxford, UK. [14]Binjiang Institute of Zhejiang University, Hangzhou, China. [15]School of Geography & Tourism, Shanxi Normal University, Xi'an, China. [16]School of Economics, Minzu University of China, Beijing, China. [17]Shaanxi Key Laboratory of Earth Surface System and Environmental Carrying Capacity, College of Urban and Environmental Science, Northwest University, Xi'an, China. [18]College of Atmospheric Sciences, Lanzhou University, Lanzhou, China. [19]Xinjiang Institute of Ecology and Geography, Chinese Academy of Sciences, Urumqi, China. [20]Key Laboratory of Desert and Desertification, Cold and Arid Regions Environmental and Engineering Research Institute, Chinese Academy of Sciences, Lanzhou, China. ✉e-mail: fhchen@itpcas.ac.cn

