## [Peer Review File · Nature Communications]

REVIEWER COMMENTS

Reviewer #1 (Remarks to the Author):

Thank you for the opportunity to read this manuscript which I think has an important focus on the adverse effects of anti-desertification measures in China. I am generally favorable to this paper and hope it will be accepted. But as I am not an expert on remote sensing myself, I find it hard to assess the methods used and to what extent the methodology is sound. As a human geographer myself, I would have liked to get to know a bit more about the region and the type of land use that is limited or banned by the anti-desertification programme. Who are excluded? How are the exclusions carried out? (fences, guards?) And what are the trends in rainfall in the area and what does the climate scenarios say about future trends? The latest IPCC report (AR6 WGII, cross-chapter paper 3) state for instance that 6% of drylands in the world are subject to drying, while 41% are subject to greening. (while this paper only mentions the first percentage...) So, greening is a much more common trend in drylands around the world than desertification. I think the paper needs to say more about current trends in this region in China as well as in the world drylands more broadly, addressing more directly the issue of greening. Otherwise, the paper reads well and makes sense to me.

Reviewer #2 (Remarks to the Author):

This short manuscript describes an impressive and expansive analysis of the environmental and economic impacts of the grain for green and grazing exclusion programs in the desertification-prone region of northern China. The authors found that these programs had limited positive impacts on environmental conditions, but reduced meat and grain production, along with incomes in local communities.

I think an analysis of these programs 10-20 years following implementation is certainly worthwhile. It is hard for me to tell how accurate most of the data are given that the environmental data are nearly all generated from remote sensing. Of course, that is necessary for any analysis at this spatial scale, but there appears to be little or no ground truthing. At least I couldn't tell from the methods section. Despite that concern the numbers are not going to change all that much and it appears that the programs cause more harm than good.

I found two things frustrating about the manuscript. First, it reads like a government report. "here's the problem and this is what we found." From the standpoint of a scientific paper there are no general concepts tested, no hypotheses or expectations against which the results can be compared. If for example only 15% of the area is improved under grain for green, is that more or less than we might expect? It is hard to decipher the numbers. In other places it was not clear what the results meant. This is particularly true from my reading regarding the amount of grain and meat being produced that seems to be well above regional needs despite a reduction in production. Perhaps I'm missing something.

The second thing that I found frustrating is that the analysis is just that, "here's the problem and this is what we found." To make this manuscript valuable and publishable, very clear recommendations need to be presented to improve upon these programs. It is not enough to say they are not working. What is needed?

What are the error estimates in the models and how are they calculated?

In summary, I think this type of assessment is valuable. I do not have enough expertise in remote sensing to judge the quality of the data and methods. I do have some sense of what makes an impactful paper and this manuscript seems to be completely focused on a management question but doesn't relate the project to bigger issues nor does it provide a strong set of recommendations based on these analysis to improve the programs being evaluated.

Reviewer #3 (Remarks to the Author):

This is an interesting and largely well written paper with a focus on China's numerous state-sponsored schemes to combat desertification. Its conclusions are important and worthy of publication in a high-ranking journal.

I have just a few, albeit important, comments:

L37 It is misleading to say the environmental and socioeconomic impacts of the practices remain unknown. There are numerous papers about the consequences. Instead of 'unknown', better to say 'unclear' and cite papers appropriately (some say the schemes have been a great success, others suggest that climate variations are more important in enhancing vegetation cover and reducing dust storm activity, for example). Relevant papers include: *Journal of Arid Environments*. 2010 1;74(1):13-22; *Land Use Policy*. 2015 Feb 1;43:42-7; *Environmental Research Letters*. 2020 Nov 18;15(11):114046.

L52-5 I'm not convinced this is a fair summation of the major countermeasures implemented, and some programmes date from much earlier than the early 2000s. See for example *Natural hazards*. 2018 92(1), pp.57-70.

L72 How have the authors done this? 'After excluding the potential contributions triggered by climate change and by vegetation physiology...'

L140 there are several other downsides to grazing exclusion programmes. See for example *Science Bulletin*. 2020 Aug 30;65(16):1405-14.

Reviewer #4 (Remarks to the Author):

1. China is one of the a few major countries who have gained vegetation coverage in recent decades. This paper intends to address the possible environmental and socioeconomic impacts of desertification combating practices in China, which is an important topic.
2. I'd suggest discussing the China desertification-combating practices in the global context, which would enhance the significance of the paper.
3. Vegetation in the studied region is also sensitive to other factors, climate for example climate, precipitation in particular. Though climate data were acquired, there is lack of climate analysis of the monitored vegetation change. Without such analysis, attributing the vegetation change to the practice programs is cursory, which lowers down the quality of the paper.
4. What land cover categories are gaining greenness, grasslands, croplands, or forests? This is important to understand the driving forces.
5. Fig. 1. It is reasonable to outline the DPR on the map so that readers know where it is located. The maps are not informative at this scale.
6. Fig. 2. A table would be more informative than this figure.
7. Fig. 3 is not meaningful without further explanation.
8. Equation (1). It is unclear min and max NDVI values are derived over what time range.

Responses to reviewer's comments:

The responses are in blue. The revisions are marked with track changes in the revised manuscript.

The comments were separated into several parts and responded to point by point.

Reviewer #1 (Remarks to the Author):

Thank you for the opportunity to read this manuscript which I think has an important focus on the adverse effects of anti-desertification measures in China. I am generally favorable to this paper and hope it will be accepted.

(1) But as I am not an expert on remote sensing myself, I find it hard to assess the methods used and to what extent the methodology is sound.

Response: Thank you for pointing this out. In the revised manuscript, we have systematically validated the data and methods used in this study and believed they are sound and led us to derive reliable results. Specifically, we (1) tested the trend consistency between the constructed fractional vegetation coverage (CD FVC) with Blended Vegetation Health (VH), Global Inventory Modeling and Mapping Studies (GIMMS3g), Satellite Pour l'Observation de la Terre (SPOT), and Moderate-resolution Imaging Spectroradiometer (MODIS) FVC products (please see Supplementary Note 3, Supplementary Fig. 8, Supplementary Table 8 and Supplementary Table 9), (2) provided accuracy and field validation details obtained from the dataset provider for the Multi-Period Land Use Land Cover Remote Sensing Monitoring Dataset for China (please see Supplementary Note 5 lines 179-185 and Supplementary Table 12), and (3) verified the simulation performance of the stepwise multiple linear regression equation for natural FVC trend driven by climate change and CO₂ fertilization based on coefficient of determination, mean absolute error and root mean square error (please see file "Manuscript without change marked.docx" lines 340-348 and Supplementary Fig. 10).

(2) As a human geographer myself, I would have liked to get to know a bit more about the

region and the type of land use that is limited or banned by the anti-desertification programme. Who are excluded? How are the exclusions carried out? (fences, guards?)

Response: We have provided detailed information about these projects in Supplementary Note 2, Supplementary Tables 1 and 6, and shown the criteria and measures to limiting or banning region and land use type in the Supplementary Table 11 (listed below). In summary, China's current ecological programs have covered 31 of 34 provinces (all DPRs and most non-DPRs also included), where most of the farmlands on slopes, desertified grasslands and forests are banned from cultivation, grazing and deforestation. To enforce such restrictions, fences or special regulatory organizations were set up for these exclusion areas to prohibit the entry of animals and humans.

Supplementary Table 11 Detailed countermeasures of ecological programs in China.

Program	Provinces included	Countermeasure	Land use type	References
P1	BJ, TJ, HE, SX, NM, LN, JL, HLJ, ZJ, AH, FJ, JX, SD, HN, HB, GD, HI, SC, YN, SN, GS, NX, QH, XJ	(1) Afforestation, (2) Closing hillsides to facilitate afforestation and sandy land protection, (3) Artificial grass planting in grassland, (4) Aeolian desertified land (grassland and farmland included) controlling, (5) Mobile sand anchoring, (6) Water-saving irrigation, and (7) Ecological resettlement	Forest, grassland, and farmland	Ref. ¹
P2	BJ, TJ, HE, SX, NM, LN, JL, HLJ, SN, GS, NX, QH, XJ	(1) Afforestation, (2) Closing hillsides to facilitate afforestation and sandy land protection, (3) Tree planting along the sides of roads, ditches, canals, and houses, (4) Forest network building for farmlands, oases, and pasture, and (5) Grass / tree planting on surfaces of mobile and semi-anchored dunes	Forest, grassland, farmland, and unused land	Ref. ^{2,3}
P3	BJ, TJ, HB, SX, NM, SN	(1) Grain for green, (2) Afforestation, (3) Grassland construction, (4) Water controlling, and (5) Ecological resettlement	Forest, grassland, desertified farmland, and sandy land	Ref. ^{3,4}
P4	NM, HLJ, JL, HI, CQ, SC, GZ, YN, HB, XZ, SX, SN, GS, NX, QH, XJ, HN	(1) Deforestation forbidding in the upper and middle reaches of Yellow River and in the upper reaches of Yangtze River, (2) Reducing timber harvesting in HLJ, JL, LN, NM, HN, and XJ, (3) Forest protection, and (4) Forest worker re-employment	Forest	Ref. ^{3,5}
P5	HLJ, JL, LN, NM, BJ, TJ, HE, HA, AH, HB, HN, JX, HI, CQ, SC, GZ,	(1) Slope treatment for farmland and desertified sandy land (slope > 25°), (2) Afforestation in mountain, hill, and sandy land with sparse vegetation, and (3) Subsidizing to farmers	Farmland and desertified sandy land (slope > 25°)	Ref. ^{3,5}

	YN, GX, SX, SN, GS, NX, QH, XJ, XZ			
P6	NM, GS, NX, XJ, XZ, QH, SC, YN, HE, SX, LN, JL, HLJ	(1) Grassland fencing and enclosing, (2) Forage-livestock balance, (3) Rotation grazing, (4) Livestock shed feeding, and (5) Grazing exclusion, subsidizing to farmers and herdsmen	Degraded grassland	Ref. ^{3,6,7}

Note: **P1:** National Sand Control Programme, aiming to combat aeolian desertification by mobile sand anchoring, afforestation, grassland management, water-saving irrigation, and ecological resettlement; **P2:** Great Green Wall Programme, aiming to anchor mobile sands and to control dust storms by afforestation; **P3:** Beijing-Tianjin Sandstorm Source Control Programme, aiming to control dust storms occurred in Beijing and Tianjin by afforestation and by returning farmland to grassland; **P4:** Natural Forest Protection Programme, afforestation and controlling deforestation; **P5:** Returning Farmland to Forests/Grassland Programme; **P6:** National Grassland Ecological Protection and Construction Programme, aiming to mitigate grassland degradation by grazing exclusion (**P6-1**), grassland fencing and closure (**P6-2**), grassland management (**P6-3**), and ecological compensation (**P6-4**).

The locations of the provinces of China are shown in Supplementary Fig. 12. Abbreviations of the provinces are established as follows: Beijing (BJ), Tianjin (TJ), Hebei (HE), Shanxi (SX), Inner Mongolia (NM), Liaoning (LN), Jilin (JL), Heilongjiang (HLJ), Shanghai (SH), Jiangsu (JS), Zhejiang (ZJ), Anhui (AH), Fujian (FJ), Jiangxi (JX), Shandong (SD), Henan (HA), Hubei (HB), Hunan (HN), Guangdong (GD), Guangxi (GX), Hainan (HI), Chongqing (CQ), Sichuan (SC), Guizhou (GZ), Yunnan (YN), Xizang (XZ), Shaanxi (SN), Gansu (GS), Qinghai (QH), Ningxia (NX), and Xinjiang (XJ).

Supplementary Fig. 12 Locations of the provinces of China covered by the programs mentioned in Supplementary Table 11.

References:

1. China National People's Congress (NPC). Interpretation of the Law of the People's Republic of China on Prevention and Control of Desertification, Chapter II Planning for Prevention and Control of Desertification. http://www.npc.gov.cn/npc/c2181/flsyywd_list.shtml (2003).
2. Wang, X. M., Zhang, C. X., Hasi, E. & Dong, Z. B. Has the Three Norths Forest Shelterbelt Program solved the desertification and dust storm problems in arid and semiarid China? *Journal of Arid Environments* 74, 13-22 (2010).
3. Bryan, B. A. et al. China's response to a national land-system sustainability emergency. *Nature* 559, 193-204 (2018).
4. PRC National Development and Reform Commission (NDRC). Planning of Beijing-Tianjin Sandstorm Source Control Program (2001-2010).

https://www.ndrc.gov.cn/fggz/fzzlgh/gjjzxgh/200709/t20070928_1196575_ext.html (2007).

5. Yin, R. & Yin, G. China's primary programs of terrestrial ecosystem restoration: initiation, implementation, and challenges. *Environ. Manage* 45, 429-441 (2010).
6. Yin, Y., Hou, Y., Langford, C., Bai, H. & Hou, X. Herder stocking rate and household income under the Grassland Ecological Protection Award Policy in northern China. *Land Use Policy* 82, 120-129 (2019).
7. Sun, J. et al. Reconsidering the efficiency of grazing exclusion using fences on the Tibetan Plateau. *Science Bulletin* 65, 1405-1414 (2020).

(3) And what are the trends in rainfall in the area and what does the climate scenarios say about future trends?

Response: Thank you for your comments. We have added current and future rainfall trends in the DPR in the revised manuscript (please see file “Manuscript without change marked.docx” lines 149-151, 153-154 and Supplementary Fig. 5 listed below). Over the past nearly 40 years, DPR experienced widespread wetting, with an overall increase trend of 2.43 mm per decade in precipitation. According to the forecast from the Coupled Model Intercomparison Project (CMIP6), such wetting trend will continue at least until 2050 CE.

Supplementary Fig. 5 Historical and future climate change trends in the DPR of China. Spatial distributions of temperature (a) and precipitation (c) trends from 1982 to 2018 and spatial distributions of temperature (b) and precipitation (d) trends from 2015 to 2050. The trend lines from 1982 to 2050 are also provided (e, f). The values in panels (b) and (d) are the average values derived under the SSP1-2.6, SSP2-4.5, and SSP5-8.5 scenarios from the CMIP6 experiments. All scenario simulations were modified by removing the errors between the CMIP6-simulated (the line referred to as Historical in panels (e,f)) and observed (the line referred to as Observation in panels (e,f)) data during the 1982-2014 reference period and the difference between the three scenarios in panels (e,f) shown since 2014. The widths of the bands in panels (e) and (f) indicate the standard deviations among the different GCMs. To compare present and future climate conditions more intuitively, the absolute climate values are replaced by anomaly values obtained based on climatic averages during the 1982 to 2015 period in panels (e) and (f). See Supplementary Note 4 for details about CMIP6 data processing.

(4) The latest IPCC report (AR6 WGII, cross-chapter paper 3) state for instance that 6% of drylands in the world are subject to drying, while 41% are subject to greening. (while this paper only mentions the first percent...) So, greening is a much more common trend in drylands around the world than desertification. I think the paper needs to say more about current trends in this region in China as well as in the world drylands more broadly, addressing more directly the issue of greening. Otherwise, the paper reads well and makes sense to me.

Response: We fully agree with you that greening is a much more common trend in the

desertification-prone region (DPR) of China as well as in drylands around the world than desertification. Following your suggestion, we have added this point, and addressed directly the role and contribution of China's current desertification combating programs within a globally greening context in the revised manuscript (please see file "Manuscript without change marked.docx" lines 147-149, 156-159).

Reviewer #2 (Remarks to the Author):

This short manuscript describes an impressive and expansive analysis of the environmental and economic impacts of the grain for green and grazing exclusion programs in the desertification-prone region of northern China. The authors found that these programs had limited positive impacts on environmental conditions, but reduced meat and grain production, along with incomes in local communities.

(1) I think an analysis of these programs 10-20 years following implementation is certainly worthwhile. It is hard for me to tell how accurate most of the data are given that the environmental data are nearly all generated from remote sensing. Of course, that is necessary for any analysis at this spatial scale, but there appears to be little or no ground truthing. At least I couldn't tell from the methods section. Despite that concern the numbers are not going to change all that much and it appears that the programs cause more harm than good.

Response: We have validated all remote sensing data. Specifically, we (1) tested the trend consistency between the constructed fractional vegetation coverage (CD FVC) with Blended Vegetation Health (VH), Global Inventory Modeling and Mapping Studies (GIMMS3g), Satellite Pour l'Observation de la Terre (SPOT), and Moderate-resolution Imaging Spectroradiometer (MODIS) FVC products (please see Supplementary Note 3, Supplementary Fig. 8, Supplementary Table 8 and Supplementary Table 9), and (2) provided accuracy and field validation details obtained from the dataset provider for the Multi-Period Land Use Land Cover Remote Sensing Monitoring Dataset for China (please see Supplementary Note 5 lines 179-185 and Supplementary Table 12). These works bring further evidence that the remote

sensing data used in this study are reliable.

(2) I found two things frustrating about the manuscript. First, it reads like a government report. “here’s the problem and this is what we found.” From the standpoint of a scientific paper there are no general concepts tested, no hypotheses or expectations against which the results can be compared. If for example only 15% of the area is improved under grain for green, is that more or less than we might expect? It is hard to decipher the numbers. In other places it was not clear what the results meant. This is particularly true from my reading regarding the amount of grain and meat being produced that seems to be well above regional needs despite a reduction in production. Perhaps I’m missing something.

Response: Thank you for pointing this out. We have thoroughly reorganized our manuscript in three aspects addressing the scientific questions and results. (1) The scientific questions were highlighted, that is, the benefit of China’s current combating desertification practices and investments remains unclear and few studies have assessed their broader impacts on sustainability (please see file “Manuscript without change marked.docx” lines 54-56). (2) We clarified the intent of this paper, that is, assessing comprehensively the environmental and economic impacts of “grain for green” and grazing exclusion practices implemented in DPR of China over the past 20 years, and provided suggestions for adapting China’s combating desertification practices and creating positive synergies to benefit livelihoods, food security, and improve the ecological environment, thereby contributing to several UN Sustainable Development Goals (please see file “Manuscript without change marked.docx” lines 61-65). (3) We clarified hypotheses or expectations against that the results can be compared, and further interpreted and discussed the results. Specifically, we indicated that China’s current desertification combating programs contributed to only 13.07% of the vegetation restoration from 2001 to 2018 in the DPR, which are much lower than the effects of climate variations and CO₂ fertilization (please see file “Manuscript without change marked.docx” lines 76-87). These current programs caused 13.4% and 24.2% mean cost in terms of foregone grain and meat production in the DPR (please see file “Manuscript without change marked.docx” lines 118-

121), and increased local financial pressure and poverty (please see file “Manuscript without change marked.docx” lines 136-143).

In addition, the DPR were the main areas for meat and grain production in China (Wei et al., 2018), and 45.4% of the household disposable income of local farmers and herders is derived directly from farming and grazing activities (please see Supplementary Table 5). Therefore, the decline in yields means a food crisis in China and local poverty, which is not only related to the local food demand. For further clarification, we have added the following sentence: “Based on the basic requirements suggested by Chinese government of 400 kilograms of grain per capita and 21 kilograms of meat per capita, these results mean that the present outputs of grain and meat in the DPR could maintain population sizes of only 59.9 million, far below the expected population of 70.6 million people in 2020” (please see file “Manuscript without change marked.docx” lines 121-124).

References:

Wei, M., Dai, J. & Sui, F. Research History of China's Reclamation. (Social Sciences Academic Press, 2018).

(3) The second thing that I found frustrating is that the analysis is just that, “here’s the problem and this is what we found.” To make this manuscript valuable and publishable, very clear recommendations need to be presented to improve upon these programs. It is not enough to say they are not working. What is needed?

Response: We appreciate your constructive suggestion. We have presented clear recommendations to improve upon these programs in the revised manuscript. According to the assessment of the spatial consequences of desertification combating programs, we suggested to terminate the land use restrictions in some DPRs, and to promote more practices such as the establishment of farmland shelterbelts, water conservation in agriculture, and the implementation of rotational grazing (please see file “Manuscript without change marked.docx” lines 186-194). Despite that an improvement of direct subsidy and a more transparent fund management system may be expected to maximize the benefits of farmers and herdsmen, we

believe that it is unsustainable under the current financial pressures of China's governments. We suggested that during the policy formulation processes of desertification combating, policy-makers should maximize the benefits of both humans and ecological environment, and create positive synergies to increase farmer and herdsman's incomes, to combat desertification and to improve the ecological environment in DPR of China. (please see file "Manuscript without change marked.docx" lines 194-202).

(4) What are the error estimates in the models and how are they calculated?

Response: We used a stepwise multiple linear regression model (SMLR) to identify the FVC natural trend driven by climate change and CO₂ fertilization, and based on this model to simulate future FVC trend. In the revised manuscript, we verified the simulation performance of the models based on the coefficient of determination (R²), mean absolute error (MAE) and root mean square error (RMSE), which are calculated as:

$$R^2 = \frac{\sum_{i=1}^n (\hat{y}_i - \bar{y}_i)^2}{\sum_{i=1}^n (y_i - \bar{y}_i)^2}$$
$$MAE = \frac{1}{n} \sum_{i=1}^n |\hat{y}_i - y_i|$$
$$RMSE = \sqrt{\frac{1}{n} \sum_{i=1}^n (y_i - \hat{y}_i)^2}$$

where, n is period length, y_i is observed FVC; \bar{y}_i is mean of y_i ; \hat{y}_i is simulated FVC. In this study, multiple regression models explained 0.53 (R²) of the change in the average in FVC and shown low values of MAE and RMSE across all land-use types, meeting the simulation requirements (please see file "Manuscript without change marked.docx" lines 340-348 and Supplementary Fig. 10 listed below).

Supplementary Fig. 10 Stepwise multiple linear regression model error. Panels (a-c) show the spatial distribution of the coefficient of determination (R^2), the mean absolute error (MAE; %), and the root mean square error (RMSE; %), respectively. The bar chart in panel (d) shows the mean statistics of R for each land use type, with the length of the error line representing the standard deviation at the pixel-scale.

(5) In summary, I think this type of assessment is valuable. I do not have enough expertise in remote sensing to judge the quality of the data and methods. I do have some sense of what makes an impactful paper and this manuscript seems to be completely focused on a management question but doesn't relate the project to bigger issues nor does it provide a strong set of recommendations based on these analysis to improve the programs being evaluated.

Response: We appreciate your attention to detail and your constructive suggestion. We have systematically verified the reliability of the remote sensing data and methods (please see the response to your comment 1 and 4). We have also clarified the scientific issues that this study addresses, the intent of this paper and hypotheses or expectations against that the results can be compared (please see the response to your comment 2), and provided detailed recommendations to improve the current programs (please see the response to your comment 3). We hope that these revisions will be satisfactory and we will be pleased to work with you to resolve any remaining issues.

Reviewer #3 (Remarks to the Author):

This is an interesting and largely well written paper with a focus on China's numerous state-

sponsored schemes to combat desertification. Its conclusions are important and worthy of publication in a high-ranking journal.

I have just a few, albeit important, comments:

(1) L37 It is misleading to say the environmental and socioeconomic impacts of the practices remain unknown. There are numerous papers about the consequences. Instead of ‘unknown’, better to say ‘unclear’ and cite papers appropriately (some say the schemes have been a great success, others suggest that climate variations are more important in enhancing vegetation cover and reducing dust storm activity, for example). Relevant papers include: Journal of Arid Environments. 2010 1;74(1):13-22; Land Use Policy. 2015 Feb 1;43:42-7; Environmental Research Letters. 2020 Nov 18;15(11):114046.

Response: We agree with your comments. We have used the word ‘unclear’ instead of ‘unknown’ and added the recommended references (please see lines 37 and 55).

(2) L52-5 I’m not convinced this is a fair summation of the major countermeasures implemented, and some programmes date from much earlier than the early 2000s. See for example Natural hazards. 2018 92(1), pp.57-70.

Response: We focus our work on the impact of “grain for green” and grazing exclusion practices on vegetation restoration, agriculture, grazing and economy in the DPR of China. These programs were implemented since the early 2000s. We did not consider earlier programs in this study such as the Great Green Wall Program launched in 1978. We have provided detailed information about all current major desertification combating programs launched in the DPR of China (please see Supplementary Note 2, Supplementary Tables 1,6 and 11). Nevertheless, we believe that these earlier programs have had an important impact on China’s ecology, and we intend to include them in future research.

(3) L72 How have the authors done this? ‘After excluding the potential contributions triggered

by climate change and by vegetation physiology...’

Response: For each pixel covered by the two programs shown in Figs.1a and 1c, we first derived the average trend in FVC driven by climate change and CO₂ fertilization (natural FVC trends) based on satellite observation and multiple linear regression simulations, and then estimated the trend components caused by programs based on the average difference between natural FVC trend and the observed actual trend with programs implementation. We have added this point in the revised manuscript (please see file “Manuscript without change marked.docx” lines 72-76, 311-340), and revised this sentence to “after exclusions of the FVC trends triggered by climate change and CO₂ fertilization (see Methods)” for a clearer expression (please see file “Manuscript without change marked.docx” line 82).

(4) L140 there are several other downsides to grazing exclusion programmes. See for example Science Bulletin. 2020 Aug 30;65(16):1405-14.

Response: Thank you for pointing this out. We have referred to your recommendation and added the negative impact of fencing for grazing exclusion on wildlife and unfenced areas (please see file “Manuscript without change marked.docx” lines 173-175).

Reviewer #4 (Remarks to the Author):

China is one of the a few major countries who have gained vegetation coverage in recent decades. This paper intends to address the possible environmental and socioeconomic impacts of desertification combating practices in China, which is an important topic.

(1) I’d suggest discussing the China desertification-combating practices in the global context, which would enhance the significance of the paper.

Response: Thank you for your constructive suggestion. In the revised manuscript, we have clarified the enlightenment of China desertification-combating practices in addressing key global issues (please see file “Manuscript without change marked.docx” lines 145-147), and discussed the role and contribution of China’s current desertification combating programs

within a globally greening context (please see file “Manuscript without change marked.docx” lines 156-159).

(2) Vegetation in the studied region is also sensitive to other factors, climate for example climate, precipitation in particular. Though climate data were acquired, there is lack of climate analysis of the monitored vegetation change. Without such analysis, attributing the vegetation change to the practice programs is cursory, which lowers down the quality of the paper.

Response: Thank you for your constructive suggestion. In the revised manuscript, we have analyzed the impact of precipitation, temperature, solar radiation intensity, wind speed, and atmospheric CO₂ concentration on vegetation restoration by stepwise multiple linear regression models (SMLRs) at pixel-scale, and determined the independent and combined contribution of these factors based on the factor coefficients in SMLRs (please see file “Manuscript without change marked.docx” lines 82, 149-153 and Fig. 2). In addition, we have predicted future FVC trends driven by climate change and CO₂ fertilization alone (please see file “Manuscript without change marked.docx” lines 153-156, Fig. 4 and Supplementary Table 7). As this paper focuses on the consequences of China’s current combating desertification practices in the DPR, we have referred to these climate-related analysis mainly in the Discussion section.

(3) What land cover categories are gaining greenness, grasslands, croplands, or forests? This is important to understand the driving forces.

Response: Thanks for your comments. In the revised manuscript, a further statistical analysis have been conducted and the results showed that 45.64% of the grasslands, 79.41% of croplands and 57.82% of forests in DPR were gaining greenness during 1982-2018 CE (please see file “Manuscript without change marked.docx” lines 69-70 and Supplementary Table 3).

(4) Fig. 1. It is reasonable to outline the DPR on the map so that readers know where it is located. The maps are not informative at this scale.

Response: Agreed. Revised as suggested.

Fig. 1 The contributions of desertification-combating practices to vegetation restoration. Spatial and statistical distributions of the contributions (%) of “grain-for-green” (a, b) and grazing exclusion (c, d) practices to vegetation restoration since 2000 and 2003, respectively. The x-axis in panels (b) and (d) corresponds to the contributions in panels (a) and (c). The scope of the DPR is marked in panels (a) and (c) with grey. For each pixel in panels (a) and (c), the average FVC trend resulting from climate change and CO₂ fertilization (natural FVC trends) is derived by satellite and multiple linear regression simulations, while the contributions of intervention practices are estimated based on the average difference between the natural FVC trend and the actual trend involving practices implementation. *Ave.α* and *Ave.β* in panels (b) and (d) are the integrated contributions of the “grain-for-green” and grazing exclusion practices, respectively. These values are calculated based on area-weighted statistics of the pixel-level contributions in Fig.1a and 1c as follows: $Ave.\alpha (\beta) = \frac{\sum [\alpha (\beta)_i \cdot area_i]}{\sum area_i}$, where $\alpha (\beta)_i$ is the vegetation restoration contribution of the two practices in *i*-th pixel involved, and *area_i* is the area of the *i*-th pixel. Noted that only regions with significant trends (passed the Mann-Kendall test at the 95% significance level) were considered in area-weighted statistics and rest were shown by the dotted box in panels (b) and (d). See Methods for more details about the identification of pixels involved in the “grain-for-green” and grazing exclusion practices, and about the calculation of pixel-level contributions.

(5) Fig. 2. A table would be more informative than this figure.

Response: The suggestion was carefully considered. We have added corresponding tables in the supplementary materials considering the length of the manuscript and its necessity (listed below).

Supplementary Table 4 Expected and actual production (tonnes) of grain and meat without the restrictions on available lands put in place by the “grain-for-green” (GG) and grazing exclusion (GE) practices in the DPR and according to the corresponding reductions (%) in production and income. In accordance with the times that these two practices were extensively launched, grain and meat production losses were estimated since 2001 and 2011, respectively.

Period	Grain production involving GG			Meat production involving GE			Income sacrifice
	Expected production	Actual production	Production loss	Expected production	Actual production	Production loss	
2001	11879640	10163032		-	-	-	
2002	13467509	11521454		-	-	-	
2003	14377161	12299661	14.45	-	-	-	9.14
2004	15201598	13004967		-	-	-	
2005	16505636	14120572		-	-	-	
2006	16218971	14009947		-	-	-	
2007	16687038	14414263		-	-	-	
2008	17769063	15348917	13.62	-	-	-	7.92
2009	17158871	14821833		-	-	-	
2010	18336576	15839134		-	-	-	
2011	18856231	16749990		2051191	1539624		
2012	20274297	18009658		2058155	1544851		
2013	22508206	19994039	11.17	2127502	1596903	24.94	17.72
2014	23337900	20731057		2223874	1669240		
2015	24359588	21638622		2237978	1679826		
2016	25128342	21482220		2181780	1669062		
2017	25469389	21773781		2119807	1621652		
2018	26574764	22718766	14.51	2057455	1573953	23.5	18.77
2019	27645832	23634422		2123305	1624328		
2020	28017946	23952542		2133431	1632075		

(6) Fig. 3 is not meaningful without further explanation.

Response: Thank you. We have added further explanations in the figure caption (please see file “Manuscript without change marked.docx” lines 204-209).

(7) Equation (1). It is unclear min and max NDVI values are derived over what time range.

Response: Thank you for your comment. We first calculated the maximum and minimum NDVI values for each year from 1982 to 2018, and then averaged these 37 pairs of values as the bare soil pixel (NDVI_{min}) and the high-density vegetation fully covered pixel (NDVI_{max}) in the study area for the period. For the sake of clarification, we have modified the relevant equations and explanations as follows (please see file “Manuscript without change marked.docx” lines 251-259):

$$CD \ FVC = \frac{NDVI - NDVI_{min}}{NDVI_{max} - NDVI_{min}}$$

(1)

$$NDVI_{\min} = \frac{1}{n} \sum_{i=1}^{i=n} NDVI_{y,\min} \quad (2)$$

$$NDVI_{\max} = \frac{1}{n} \sum_{i=1}^{i=n} NDVI_{y,\max} \quad (3)$$

where $NDVI_{\max}$ (or $NDVI_{y,\max}$) and $NDVI_{\min}$ (or $NDVI_{y,\min}$) correspond to the NDVI values for a surface with a fully covered high-dense vegetation and for bare soils, respectively. To reduce the uncertainty and randomness in determining extreme NDVI values¹, we first selected the NDVI values ($NDVI > 0$) at the 5% and 95% percentiles as annual $NDVI_{y,\min}$ and $NDVI_{y,\max}$ for each year, respectively, and then calculated the multi-year average $NDVI_{y,\min}$ and $NDVI_{y,\max}$ values as the bare soil pixel ($NDVI_{\min}$) and the high-density vegetation fully covered pixel ($NDVI_{\max}$) in the study area for the period of 1982 to 2018.

References:

1. Di Traglia, F. & Gerlach, R. Portfolio selection: An extreme value approach. *J. Bank. Financ.* 37, 305-323 (2013).

REVIEWERS' COMMENTS

Reviewer #1 (Remarks to the Author):

I think the authors have responded convincingly to the reviewer reports and that the manuscript is ready for acceptance

Reviewer #2 (Remarks to the Author):

Thank you for your careful and thoughtful revisions. I'm satisfied with this version of the manuscript. The analysis does come across as "environment vs economics" which is unfortunate because it perpetuates that myth that one comes at the cost of the other. The region is getting greener but the people are now poorer.

Only one tiny suggestion is that you include an economic component to the last sentence of your abstract

Reviewer #3 (Remarks to the Author):

The authors have adequately dealt with all my comments except one. I made the point that some programmes date from much earlier than the early 2000s. I cannot see how they can ONLY consider programmes implemented since the early 2000s because the earlier programmes are likely still to have a residual effect. This may be difficult to quantify, certainly, but they cannot simply ignore the programmes pre-2000. They must be mentioned in the paper. Doubly so because I see from *Natural Hazards*. [2018 92(1), pp.57-70] that the Great Green Wall, which was launched in 1978, is not scheduled to be complete until 2050!

They should also now cite the latest paper on this issue:
Wu, C., Lin, Z., Shao, Y., Liu, X. and Li, Y., 2022. Drivers of recent decline in dust activity over East Asia. *Nature Communications*, 13(1), pp.1-10.

Reviewer #4 (Remarks to the Author):

I think the authors did a great job addressing reviewer's comments with detailed responses. In particular, my comments have been satisfactorily addressed.

Responses to reviewer's comments:

The responses are in blue. The revisions are marked in red in the revised manuscript.

The comments were separated into several parts and responded to point by point.

Reviewer #1 (Remarks to the Author):

I think the authors have responded convincingly to the reviewer reports and that the manuscript is ready for acceptance

Response: Thank you for your positive comments.

Reviewer #2 (Remarks to the Author):

Thank you for your careful and thoughtful revisions. I'm satisfied with this version of the manuscript. The analysis does come across as "environment vs economics" which is unfortunate because it perpetuates that myth that one comes at the cost of the other. The region is getting greener but the people are now poorer.

Only one tiny suggestion is that you include an economic component to the last sentence of your abstract.

Response: Thank you for pointing this out. We have modified the sentence to “China needs to adapt its environmental programmes to address the potential impacts of future climate change and create positive synergies to combat desertification and improve the economy in this region.” (please see lines 42-43).

Reviewer #3 (Remarks to the Author):

The authors have adequately dealt with all my comments except one. I made the point that some programmes date from much earlier than the early 2000s. I cannot see how they can ONLY consider programmes implemented since the early 2000s because the earlier programmes are likely still to have a residual effect. This may be difficult to quantify, certainly, but they cannot simply ignore the programmes pre-2000. They must be mentioned in the paper.

Doubly so because I see from Natural Hazards. [2018 92(1), pp.57-70] that the Great Green Wall, which was launched in 1978, is not scheduled to be complete until 2050!

They should also now cite the latest paper on this issue:

Wu, C., Lin, Z., Shao, Y., Liu, X. and Li, Y., 2022. Drivers of recent decline in dust activity over East Asia. Nature Communications, 13(1), pp.1-10.

Response: Thank you for your constructive suggestions. We have mentioned the programmes implemented before 2000s such as the Great Green Wall Program (launched in 1978) having influence on vegetation condition, and cited papers you suggested (please see lines 48-51, 297-300). Nevertheless, this work aims to address the issues about ecological and economic trade-offs arising from farmland and grassland restrictions in combating desertification, which is overwhelmingly related to the “grain-for-green” and grazing exclusion practices. This is the reason why we focus on these two programmes implemented since 2000s. We have explained this in the method section (please see lines 297-300).

In addition, given the accurate identification of the areas covered by “grain-for-green” and grazing exclusion practices, the impact of residual effects of these earlier practices such as the Great Green Wall Program on the assessment results have also been considered (please see lines 292-297).

Reviewer #4 (Remarks to the Author):

I think the authors did a great job addressing reviewer’s comments with detailed responses. In particular, my comments have been satisfactorily addressed.

Response: Thank you for your positive comments.